# Recycling Waste Electrical and Electronic Equipment (WEEE) and the Management of Its Toxic Substances in Taiwan—A Case Study

**DOI:** 10.3390/toxics8030048

**Published:** 2020-07-07

**Authors:** Wen-Tien Tsai

**Affiliations:** Graduate Institute of Bioresources, National Pingtung University of Science and Technology, Pingtung 912, Taiwan; wttsai@mail.npust.edu.tw; Tel.: +886-8-7703202

**Keywords:** waste electrical and electronic equipment, recycling, toxic substance, regulation, Taiwan

## Abstract

In the past two decades, the waste electrical and electronic equipment (WEEE) management has become an important environmental issue internationally because it contained hazardous substances like heavy metals and brominated flame retardants. Moreover, some valuable substances were used in the electrical and electronic products, thus representing a circular industry for recycling of WEEE. Therefore, the Taiwan government formulated a legal WEEE recycling system since 1998 in response to the international trends of sustainable waste management and extended producer responsibility (EPR). This article adopted the national statistics in Taiwan regarding the online reporting amounts of collected WEEE since it has been officially designated as one of the mandatory recyclable wastes. Furthermore, the regulatory measures were addressed to update the status and subsidiary fee rates of WEEE recycling in Taiwan. In addition, this article also put emphasis on the regulations governing the toxic chemical substances contained in the WEEE. It showed that the average annual recycling amounts of home electronic appliances, information technology products and lighting in Taiwan during the 2017–2018 were around 117,000, 18,000 and 4500 metric tons, respectively. It was also indicated that the current WEEE recycling market in Taiwan has become saturated, reflecting the regulatory promulgation and promotional measures successfully. In response to the Stockholm Convention on persistent organic pollutants (POPs) and the Minamata Convention on Mercury, the Taiwan government declared some brominated flame retardants and heavy metals (i.e., mercury and cadmium) as a “toxic chemical substance” under the Toxic and Concerned Chemical Substance Control Act (TCCSCA), which shall be prohibited to use in the preparation of electrical and electronic equipment (EEE) since 1 January 2016. Through the central governing authority, local governments, and private recyclers in Taiwan, the successful WEEE recycling system not only reduce the pressure on sanitary disposal systems, but also prevent the chemical hazards from solid waste incineration systems. More significantly, the WEEE recycling in Taiwan echoed the United Nations (UN) Agenda 2030 for sustainable development goals.

## 1. Introduction

Over the past decades, municipal solid waste (MSW) generation has increased at a significant rate, which was linked to economic growth and living level. As electrical and electronic equipment often contained a large number of hazardous substances such as heavy metals (e.g., mercury), brominated flame retardants (e.g., polybrominated diphenyl ethers) and other substances, its waste electrical and electronic equipment (WEEE) in the MSW could cause serious environmental pollution and human health problems from disassembling/pulverizing facilities, incineration facilities, sanitary landfills and illegal dumping sites [1,2,3,4,5,6,7,8]. More noticeably, these disposed or incinerated wastes may imply the depletion of valuable resources without recycling or reuse. In response to the international trends on sustainable development in the 1990s, the resulting discards have been implemented in the 3R (Reduce, Reuse, and Recycle) frameworks by rethinking them as valuable resources [9,10,11,12,13,14]. In this regard, the rapidly growing stream of WEEE continued to be a serious challenge because it is directly related to illegal exports or imports around the world. As the world becomes increasingly interconnected and as electrical and electronic products (i.e., personal computers, gadgets, digital cameras, and cell phones) are quickly devalued and become obsolete due to the product cost-down and functional advancement, the expansion of the WEEE stream will become an even greater challenge for MSW management in the developing and developed countries [15,16,17,18,19,20,21,22,23,24,25,26,27,28,29,30,31].

As the amount of WEEE generated around the world has reached over 40 million metric tons annually [32], with the major contributors coming from the United States, European Union (EU) and Asia, the EU thus passed legislations to restrict the use of hazardous substances in electrical and electric equipment (“Restriction of Hazardous Substances Directive”, RoHS 2002/95/EC), which was adopted in February 2003. This Directive was to promote the collection and recycling of such equipment, and also reduce the amount of hazardous substances dispersed through recycling operations, especially shredding residues that could be contaminated by hazardous materials. Another Directive 2002/96/EC on WEEE is a key element of the EU’s environmental policy on waste management and addresses a particularly complex waste flow. The objectives of this Directive include:(i)Diverting WEEE from sanitary landfills and MSW incinerators to eco-friendly reuse, recycling, and other forms of recovery;(ii)Preserving resources, raw materials, and energy;(iii)Encouraging manufacturers’ responsibility;(iv)Integrating national measures on WEEE management, which put in place common minimum standards for treatment.

Since the implementation of the EU Directive on WEEE, this waste policy has become a rising environmental issue in the developed and developing countries, especially in Asian countries [18].

In order to echo with the international trends regarding the sustainable waste management and the extended producer responsibility (EPR) [16,33], the central competent authority (i.e., Environmental Protection Administration (EPA)) in Taiwan promulgated the resource recycling system under the authorization of the Waste Management Act since 1998. More significantly, the EPA began promoting the 4-in-1 Program, which claimed the responsible enterprises to pay a recycling fee to the Recycling Management Fund. The Fund’s money will be mostly used as an incentive to integrate local governments, communities, and recycling enterprises for resource recycling promotion. Mainly due to its value for recycling and hazardous substances contained, the EPA continuously declared the regulated recyclable waste items. The status of WEEE recycling in Taiwan has been reviewed by the previous study [30] and other reports [34,35]. These recycled WEEE articles were carried out by the registered recycling enterprises, which can be subsidized by the special recycling fund. In addition, another resource recycling system in Taiwan for usable yet outdated products was conducted by the civil non-profit organizations that may be not registered as the recycling enterprises. Among them, the Tzu Chi, one of the Buddhist groups, may be the most famous non-profit recycling systems.

Very little work on the overview of WEEE recycling and its regulatory concerns about toxics contained in the WEEE in Taiwan was available in the published literature. This case report aimed to review the regulations governing WEEE recycling in Taiwan. Furthermore, the updated status of WEEE recycling and their fee rates for the responsible enterprises were also addressed in the study. Finally, this paper interactively presented the regulatory concerns about toxics contained in the WEEE because these heavy metals and/or brominated flame retardants were restricted on use by the international initiatives or conventions like the Stockholm Convention on persistent organic pollutants (POPs) [36] and the Minamata Convention on Mercury [37]. 

## 2. Overview of Regulatory System for WEEE Recycling in Taiwan

In Taiwan, the current waste management policy focuses on the promulgations of waste generation prevention and resource recycling for its exhausted or consumed equipment, which meet the sustainable development goals on and sustainable material flow and low-carbon society. The legal system for WEEE recycling in Taiwan is based on the Waste Management Act (also called Waste Disposal Act) [38,39]. According to the requirements by the Act, the principle of extended producer responsibility (EPR) was adopted to promote WEEE recycling by the subsidization and the reduced fee rates for the eco-friendly electrical and electronic equipment. The registered recycling enterprises that meet stringent standards can be subsidized by the central competent authority. This section may be divided by subheadings for providing an overview of regulatory system for WEEE recycling and its updated fee rates in Taiwan.

### 2.1. Act Governing WEEE Recycling

The Waste Management Act, recently revised on 14 June 2017, designated WEEE as one of the regulated recyclable wastes since 1998 due to the promulgation of the 4-in-1 Recycling Program [40,41,42]. Under the Article 15 of the Act, the regulated recyclable wastes are those that could cause the concerns about serious environmental pollution and also possess the following characteristics:(i)Being difficult to disposal of;(ii)Containing components or substances that do not readily decompose in the natural environment;(iii)Containing hazardous substances;(iv)Being valuable for recycling.

Obviously, the WEEE is in accordance with the above-mentioned characteristics. The EPA continuously listed certain WEEE items since 1998 as the regulated recyclable wastes [27,30,35,40], including home electric appliances, information and communication technology (ICT) products and lighting. The 4-in-1 Recycling Program integrated community residents, responsible and recycling enterprises, local governments, and the non-profit recycling fund (the Recycling Management Fund) to carry out the recycling of the regulated recyclable wastes effectively. Figure 1 showed the simplified operation mode of the 4-in-1 Recycling Program [41]. The Recycling Management Fund (RMF) plays a key role in the performance of the 4-in-1 Program because it annually collects around USD 30 million from the responsible enterprises (manufacturers and importers of regulated products). The recycling fee is used to subsidize the registered private collection and treatment enterprises. These fees were mainly used to subsidize the recycling enterprises, provide the rewards or incentives for the promotion of recycling system, and also pay the auditing and certification organizations for the execution expense.

### 2.2. Fee Rates for WEEE Recycling

As mentioned above, the responsible enterprises shall register with the EPA and also pay recycling fees to an officially assigned bank based on the manufacturing volume or imported volume reported to the customs according to the fee rates, which are approved by the RFMB. The Fee Rate Review Committee, which is established by the EPA, shall perform the fee rate reviews for the regulated recyclable wastes based on their materials, volumes, weights, impacts on the environment, recyclable values, recycling and disposal costs, recycling and disposal ratios, auditing and collection costs, financial conditions of the Fund, and other relevant factors. The Committee shall submit the review to the RFMB for approval and official announcement. Table 1, Table 2 and Table 3 summarized the current recycling fee rates of home electrical appliances, IT products and lighting, respectively. The recyclable home electrical appliances included television (TV) sets, refrigerators, washing machines, air conditioners, and electric fans. By contrast, the recyclable IT products covered portable computers, monitors, mainboards, hard drives, cases, power supplies, printers, and keyboards. In order to encourage the responsible enterprises to make eco-friendly or green-mark products, Table 1 and Table 2 also showed the reduced recycling fee rates of home electrical appliances and IT products, which shall be certified by the central competent authorities (i.e., the EPA, and the Ministry of Interior, and the Ministry of Economic Affairs). On the other hand, the reduced recycling fee rates for the liquid crystal display (LCD) TV set with larger than 27 inches were effective at different implementation stages (seen in Table 1). This measure was intended to give the affected enterprises for sufficient adaption time and also to provide economic incentives for the responsible enterprises.

## 3. Status of WEEE Recycling in Taiwan

### 3.1. Status of WEEE Recycling Enterprise

According to the Article 18 of the Waste Management Act, the recycling enterprises shall register with the EPA and declare their recycling volumes and related operation records regularly. Their recycling operations by the recycling enterprises must be in accordance with the official standard (“Standards of Storage, Clearance and Treatment Methods and Facilities for Recycling Waste Electrical and Electronic Appliances, and IT Equipment”), which stated the standards of collection, storage, clearance, and treatment methods and facilities for the designated items listed in Table 1, Table 2 and Table 3. According to the Standards, the collected WEEE will be sent to the registered treatment enterprises. These recyclers further adopted proper machines and procedures to disassemble, pulverize and separate the component materials or pieces from the WEEE. In order to prevent the hazardous substances (especially for mercury and cadmium) from discharging from the dismantling processes, the WEEE (e.g., fluorescent tube, cathode ray tube and liquid crystal display) must be treated by the non-destruction methods under airtight conditions. Due to the subsidized supports by the Recycling Management Fund, the registered WEEE recycling enterprises, summarized in Table 4 [41], have formed a circular economy industry in Taiwan.

### 3.2. Status of WEEE Recycling

In Taiwan, WEEE can be collected through implementing agencies, including private recycling systems, vendors, local government municipal solid waste (MSW) collection crews, individual recyclers, and registered collection facilities, or directly from the manufacturers and importers (i.e., responsible enterprises). According to the government regulations, the collected WEEE will be sent to the registered treatment facilities. Due to the official declaration based on the unit of set before 2016, it was not available for the statistics on the amounts of recycled home electrical and electronic appliances and IT products by weight except for waste lighting. Regarding the recycling achievements for the regulated WEEE over the last two years (from 2017 to 2018), Table 5 summarized the statistics on the amounts of recycled WEEE in Taiwan [39]. In brief, the average annual recycling volume of waste electrical and electronic appliances was around 117,000 metric tons; the average annual recycling volume of waste IT equipment was 18,000 metric tons; and the average annual recycling volume of waste lighting was 4500 metric tons. 

As the recycling system of waste lighting resources was complete and the EPA have sustainably held the recycling propagation of waste lighting, the recycling volume of waste lighting grew from 524 metric tons in 2002 to 4555 metric tons in 2018 (seen in Figure 2); a total of 1954 kg of mercury was recycled. Figure 2 showed the certified recycling volume of waste lighting from 2002 to 2018 [39]. The certified qualities of recyclable lighting indicated a declining trend since 2013. This shift change could be attributed to the mature recycling market [42]. On the other hand, the use of light emitting diode (LED) lighting was extensively used in Taiwan due to the official promotion by its efficient energy saving. Table 6 summarized the implementation date for declaring lighting products as mandatory recycling items in Taiwan.

## 4. Regulatory Concerns about Toxics Contained in the WEEE

As the electrical and electronic equipment must meet the performance specifications or standards, they often contained some hazardous substances, including heavy metals (i.e., mercury, cadmium, lead, etc.) and brominated flame retardants (e.g., polybrominated diphenyl ethers, hexabromobiphenyl, hexabromocyclododecanes, etc.). In this regard, WEEE is generally designated as hazardous waste because it may pose significant human health and environmental pollution risks if improperly managed. As mentioned above, in Taiwan, the most important initiative that would bear on hazardous waste was the Basel Convention on the Control of Transboundary Movements of Hazardous Wastes and their Disposal, which came into force in 1992. Subsequently, the European Union WEEE Directive entered into force in 2003, which set targets for the collection, recovery, and recycling of WEEE covering 10 categories. On the other hand, its complementary directive is the Restriction of Hazardous Substances (RoHS), which required the substitutions of heavy metals (i.e., lead, mercury, cadmium, hexavalent chromium) and flame retardants (i.e., polybrominated biphenyls and polybrominated diphenyl ethers) in new electrical and electronic equipment from 1 July 2006 for the purpose of preventing hazardous waste generation. In addition, the Stockholm Convention on persistent organic pollutants (POPs), which came into effect on 17 May 2004, decided to blanket certain polybrominated biphenyls and polybrominated diphenyl ethers into the POPs in 2009 [36]. On the other hand, refrigerants (i.e., CFC-11, HFC-134a) must be recycled and reused while processing waste refrigerators and air conditioners according to the regulations.

In order to comply with the national initiatives governing the restrictive uses of hazardous substances in the electrical and electronic equipment, the Bureau of Standards, Metrology and Inspection (one of the implementing agencies under the Ministry of Economic Affairs) promulgated the Ordinance CNS 15,663 (i.e., “the Guidance to Reduction of the Restricted Chemical Substances in Electrical and Electronic Equipment”) on 30 July 2013. Table 7 showed the reference values of percentage content (i.e., maximal levels) for the restricted chemical substances. Furthermore, these hazardous substances have been listed as the “toxic chemical substance” by the central competent authority (i.e., EPA) under the Toxic and Concerned Chemical Substance Control Act (TCCSCA). This section will summarize the main management requirements based on the TCCSCA.

### 4.1. Regulatory Concerns about Heavy Metals Contained in the WEEE

As listed in Table 8, mercury (Hg) and cadmium (Cd) have been designated as toxic chemical substances under the TCCSCA. According to the Article 11 of the TCCSCA, the EPA shall announce the control levels (composition standards) and graded threshold handling quantities for toxic chemical substances. Herein, the toxicity classification is subject to the regulatory definition as follows:(i)Class 1 toxic chemical substances: those substances that are not prone to decompose in the environment or that pollute the environment or endanger human health due to bioaccumulation, bioconcentration or biotransformation.(ii)Class 2 toxic chemical substances: those substances that can cause tumors, infertility, teratogenesis, genetic mutations or other chronic diseases.(iii)Class 3 toxic chemical substances: those substances that can endanger human health or the lives of biological organisms immediately upon exposure.

It is well known that the accumulation of mercury in the body over a long period can attack the target organs (e.g., brain) or nervous system [1]. To prevent the release of mercury into the environment, mercury was listed by the EPA as the Class 1 toxic chemical substance on 7 December 1991. Subsequently, the EPA revised the usage and restrictions on mercury by several times to reduce the exposure risk to Hg-containing products when they were disposed of and treated. On 5 July 2019, the EPA further declared the prohibitive manufacturing for Hg-containing commodities, including button cells, switches and relays, lighting fixtures, fluorescent tubes, cosmetics, as well as non-electronic measuring instruments (i.e., barometers, hygrometers, pressure gauges, thermometers and blood-pressure meters), since 1 January 2021 because the Minamata Convention on Mercury took effect on 16 August 2017 [37].

### 4.2. Regulatory Concerns about Organic Toxics Contained in the WEEE

As described above, the electrical and electronic equipment often contained polychlorinated biphenyls, and flame retardants like polybrominated biphenyls and polybrominated diphenyl ethers. These persistent organic compounds pose some human health and environmental risks due to their high bioaccumulation and bioconcentration factors, thus listed as POPs in the Stockholm Convention since 2009. In order to be accordance with the international initiatives, the EPA declared some flame retardants (decabromodiphenyl ether not declared) as the Class 1 toxic chemical substances under the TCCSCA on 25 August 2014. In addition, these declared flame retardants shall be prohibited to use in the preparation of electrical and electronic equipment since 1 January 2016. On 5 March 2019, the EPA further declared decabromodiphenyl ether as the Class 1 and Class 2 toxic chemical substance based on the revised Stockholm Convention. Table 9 listed brominated flame retardants (BFRs) designated as toxic chemical substances under the TCCSCA.

## 5. Conclusions and Prospects

In recent years, the public has become more concerned about the environment pollution and human health impacts as a result of waste electrical and electronic equipment (WEEE) recycling, including home electronic appliances, information technology products and lighting. Under the authorization of the Waste Management Act, the Taiwan government thus promulgated the 4-in-1 Program for WEEE recycling since 1998 based on the international trends of sustainable waste management and extended producer responsibility (EPR). With the implementation of WEEE recycling in Taiwan, the average annual recycling volume of WEEE was around 140,000 metric tons, and the collection rate for 2018 was over 60%. This approach not only reduced the amount of WEEE requiring incineration and disposal, but also recycled the valuable resources from WEEE. Due to the toxicity and environmental persistence, some brominated flame retardants and heavy metals (i.e., mercury and cadmium) shall be prohibited to use in the preparation of electrical and electronic equipment because they have been recently declared as the “toxic chemical substance” under the Toxic and Concerned Chemical Substance Control Act (TCCSCA). 

To raise the WEEE recycling performance, the Taiwan EPA shall adopt further promotional measures, including an increase in the number of regulated WEEE items, the recycling fee rate of the green-mark ICT products enhanced, the planning of differential subsidy fee improved, and the expanded scale of recycling enterprises by merge. In summary, the zero waste and total resource recycling towards the green supply chain and circular economy will be prospective in the Taiwan’s WEEE recycling system.

## Figures and Tables

**Figure 1 toxics-08-00048-f001:**
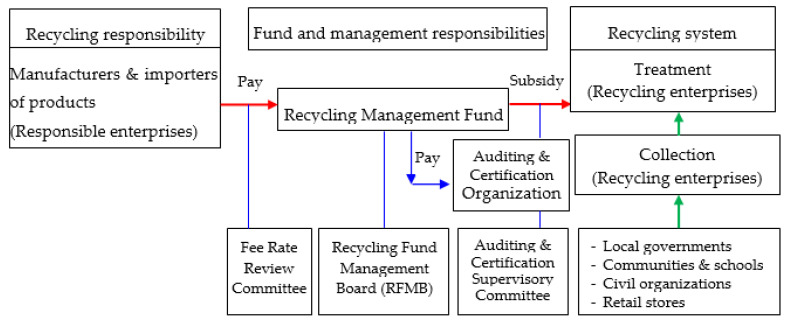
Simplified operation mode of the 4-in-1 Recycling Program in Taiwan.

**Figure 2 toxics-08-00048-f002:**
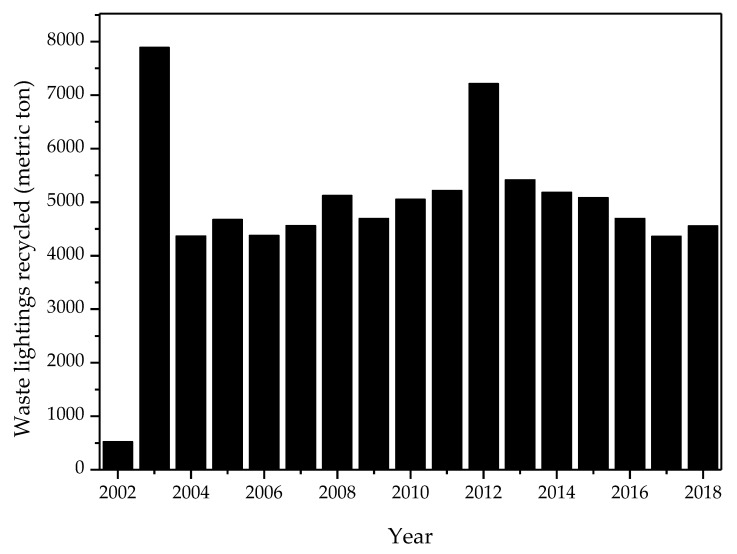
Amount of waste lighting recycled during 2003–2018 in Taiwan [39].

**Table 1 toxics-08-00048-t001:** Current recycling fee rates of recyclable home electrical appliances in Taiwan.

Items			Recycling Fee Rate (TWD/set) ^a^		Effective Date
			Regular	Green-Mark Products	
Television (TV) sets	Non-LCD	>27 in.	397	378	1 March 2020
		≤27 in.	277	264	
	LCD	>27 in.	275	262	
		≤27 in.	127	121	
	Non-LCD	>27 in.	397	378	1 March 2021
		≤27 in.	277	264	
	LCD	>27 in.	317	302	
		≤27 in.	127	121	
	Non-LCD	>27 in.	397	378	1 March 2021
		≤27 in.	277	264	
	LCD	>27 in.	275	262	
		≤27 in.	127	121	
Refrigerator		>250 L	588	567	1 March 2020
		≤250 L	444	378	
Washing machine			357	304	1 March 2020
Air conditioner			297	253	1 March 2020
			353	300	1 March 2021
			429	365	1 March 2022
Electric fan	>12 in.		34	29	1 January 2015
	≤12 in.		19	16	

^a^ USD/TWD = 30 (2020).

**Table 2 toxics-08-00048-t002:** Current recycling fee rates of recyclable IT products in Taiwan.

Items		Recycling Fee Rate (TWD/set) ^a^		Effective Date
		Regular	Green-Mark Products	
Portable PC	Notebook	39	34	1 March 2020
	Tablet	25.3	22	
Monitor	Non-LCD	127	108	
	LCD	127	108	
Mainboard		47.6	40.5	
Hard disk		47.6	40.5	
Case		7.9	6.6	
Power supply		7.9	5.6	
Printer	Ink-jet type	175	167	
	Laser type	193	184	
	Dot-matrix type	188	179	
Keyboard		14	12	

^a^ USD/TWD = 30 (2020).

**Table 3 toxics-08-00048-t003:** Current recycling fee rates of recyclable lighting in Taiwan ^a^.

Items	Recycling Fee Rate (TWD/kg) ^a^		Effective Date
Fluorescent light tube (straight)	41		1 January 2017
Circular fluorescent bulb, self-ballasted fluorescent bulb, compact fluorescent bulb, incandescent bulb (>2.6 cm I.D.), high intensity discharge (HID), other mercury-containing lamp	31		
Cold cathode fluorescent lamp (CCFL)	26.9		
Magnetic induction lamp (MIL)	25.7		
Straight tube, circular tube, self-ballasted, compact light emitting diode (LED)	25.8		1 January 2017–30 June 2020
	25.8	Recycling ≥ 90%	1 July 2020
	23.9	90 > Recycling ≥ 80%	
	22.0	Recycling < 80%	

^a^ USD/TWD = 30 (2020).

**Table 4 toxics-08-00048-t004:** Registered E-waste collection and treatment facilities in Taiwan ^a^.

WEEE Type	Registered Recycling Enterprise	
	Collection	Treatment
Home electronic appliances	233	18
Information technology (IT) products	232	23
Lighting	169	3

^a^ Source [41].

**Table 5 toxics-08-00048-t005:** Statistics on the amounts of recycled WEEE in Taiwan.

Year	Home Electrical and Electronic Appliances (Metric Ton)	IT Products (Metric Ton)	Lighting (Metric Ton)
2017	107,329	19,934	4361
2018	127,237	16,460	4555

**Table 6 toxics-08-00048-t006:** Implementation date that lighting was listed as one of mandatory recyclables in Taiwan.

Implementation Date	Lighting Products Recycled
1 January 2002	Fluorescent light tube (straight type)
1 July 2007	Circular fluorescent bulb, self-ballasted fluorescent bulb, compact fluorescent bulb, incandescent bulb (>2.6 cm I.D.)
1 March 2014	Cold cathode fluorescent lamp (CCFL), magnetic induction lamp (MIL), other mercury-containing lamps
1 January 2017	Light emitting diode (straight tube, circular tube, compact, and built-in ballast type)

**Table 7 toxics-08-00048-t007:** The reference values of percentage content for the restricted chemical substances promulgated by the Ordinance CNS 15,663 in Taiwan.

Restricted Substance	Substance to Be Calculated	Reference Values of Percentage Content (wt%)
Lead and its compounds	Lead (Pb)	0.1
Mercury and its compounds	Mercury (Hg)	0.1
Cadmium and its compounds	Cadmium (Cd)	0.01
Chromium (VI) compounds	Hexavalent chromium (Cr^+6^)	0.1
Polybrominated biphenyls	Polybrominated biphenyls (PBB)	0.1
Polybrominated diphenyl ethers	Polybrominated diphenyl ethers (PBDE)	0.1

Remark note: The reference value of mercury content for each single-capped fluorescent lamp shall be less than or equal to 5 mg, or in accordance with the regulations by the EPA. The exemption conditions (or articles) for the reference value of content refer to the Annex D of the Ordinance.

**Table 8 toxics-08-00048-t008:** Heavy metals designated as toxic chemical substances under the TCCSCA.

Heavy Metal	Molecular Formula	CAS No.	Control Level ^a^ (wt%)	Threshold Handling Quantity (kg)	Toxicity Classification
Mercury	Hg	7439-97-6	95	50	1
Cadmium	Cd	7440-43-9	95	500	2, 3

^a^ The control level is officially announced by the central competent authority when handling the concentration of toxic chemical substance greater than the standard.

**Table 9 toxics-08-00048-t009:** Brominated flame retardants (BFRs) designated as toxic chemical substances under the TCCSCA.

Brominated Flame Retardant	Molecular Formula	CAS No.	Control Level (wt%)	Threshold Regulation Quantity (kg)	Toxicity Classification
Decabromodiphenyl ether	C_12_Br_10_O	1163-19-5	1	50	1, 2
Pentabromodiphenyl ether	C_6_Br_3_H_2_-O-C_6_Br_2_H_3_	32534-81-9	1	50	1
2,2′,4,4′-tetrabromo- diphenyl ether (BDE-47)	C_12_H_6_Br_4_O	40088-47-9	1	50	1
2,2′,4,4′,5,5′-Hexabromo- diphenyl ether (BDE-153)	C_12_H_4_Br_6_O	68631-49-2	1	50	1
2,2′,4,4′,5,6′-Hexabromo-diphenyl ether (BDE-154)	C_12_H_4_Br_6_O	207122-15-4	1	50	1
2,2′,3,3′,4,5′,6-Heptabromo- diphenyl ether(BDE-175)	C_12_H_3_Br_7_O	446255-22-7	1	50	1
2,2′,3,4,4′,5′,6-Heptabromo- diphenyl ether (BDE-183)	C_12_H_3_Br_7_O	207122-16-5	1	50	1
Hexabromobiphenyl	C_12_H_4_Br_6_	36355-01-8	1	50	1
1,2,5,6,9,10-Hexabromo- cyclododecane (HBCD)	C_12_H_18_Br_6_	3194-55-6 25637-99-4	1	50	1
α-Hexabromocyclo- dodecane	C_12_H_18_Br_6_	134237-50-6	1	50	1
β-Hexabromocyclo- dodecane	C_12_H_18_Br_6_	134237-51-7	1	50	1
γ-Hexabromocyclo- dodecane	C_12_H_18_Br_6_	134237-52-8	1	50	1

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
