# Peer review of "Recycling Waste Electrical and Electronic Equipment (WEEE) and the Management of Its Toxic Substances in Taiwan—A Case Study"

_toxics, 2020, doi:10.3390/toxics8030048_

Round 1
Reviewer 1 Report
Here my comments:
Line 14. For recycling WEEE, it must be recycling of WEEE.
Line 30: electrical and electronic equipment: add in the end of this sentence (EEE)
Line 30: 1 Jan. 2016: write Jan completely January.
Line 42 its waste (WEEE): its waste of electric and electronic equipment (WEEE)
Line 48: the rapidly-growing stream of WEEE (E-waste or electronic waste): eliminate (E-waste or electronic waste)
Line 120: The Recycling Management Fund add (RMB)
Line 141-142: which are approved by the Recycling Fund Management Board (RFMB). Eliminate the Recycling Fund Management Board is already mentioned before.
Line 188: Due to the official declaration based on the unit of set, it not. It is missing is between it and not.
Line 218: (Waste Electrical and Electronic Equipment) is already mentioned before keep WEEE.
Line 220- 226: what about CFC in refrigerators: (chlorofluorocarbon). It is important to mention this toxic organic compounds.
Line 267: in Regulatory concerns about organic toxics contained in the WEEE, the chlorine content in WEEE poses some problems as bromine.
Author Response
Q1. Line 14. For recycling WEEE, it must be recycling of WEEE.
Reply: It has been corrected to make it clear.
Q2. Line 30: electrical and electronic equipment: add in the end of this sentence (EEE).
Reply: It has been corrected to make it clear.
Q3. Line 30: 1 Jan. 2016: write Jan completely January.
Reply: It has been corrected to make it clear.
Q4. Line 42 its waste (WEEE): its waste of electric and electronic equipment (WEEE).
Reply: It has been corrected to make it clear.
Q5. Line 48: the rapidly-growing stream of WEEE (E-waste or electronic waste): eliminate (E-waste or electronic waste).
Reply: It has been corrected to make it clear.
Q6. Line 120: The Recycling Management Fund add (RMB).
Reply: It has been added to make it clear.
Q7. Line 141-142: which are approved by the Recycling Fund Management Board (RFMB). Eliminate the Recycling Fund Management Board is already mentioned before.
Reply: It has been corrected to make it clear.
Q8. Line 188: Due to the official declaration based on the unit of set, it not. It is missing is between it and not.
Reply: It has been corrected to make it right.
Q9. Line 218: (Waste Electrical and Electronic Equipment) is already mentioned before keep WEEE.
Reply: It has been corrected to make it consistent.
Q10. Line 220- 226: what about CFC in refrigerators: (chlorofluorocarbon). It is important to mention this toxic organic compounds.
Reply: As suggested by the reviewer, the information about the CFC in refrigerators has been incorporated into the end of the paragraph.
“……On the other hand, refrigerants (i.e., CFC-11, HFC-134a) must be recycled and reused while processing waste refrigerators and air conditioners according to the regulations.”
Q11. Line 267: in Regulatory concerns about organic toxics contained in the WEEE, the chlorine content in WEEE poses some problems as bromine.
Reply: It has been corrected to make it clear.

Reviewer 2 Report
In this manuscript, the author introduced the case study of the recycling of waste electrical and electronic equipment (WEEE) and its management of toxic substances in the Taiwan government.
I think it will be useful to some extent for the readers.
However, this manuscript only introduces the measures of the Taiwan government, and does not suggest the academic significance of social science or the improvement of the system.
It will be necessary to explain what is different from the text of the government policy introduction. Authors should also consider and suggest future improvements to WEEE.
p.3 l.120
What is the 4-in-1 Recycling program? What does 4 or 1 represent? For example, what part is better than overseas accuracy?
p.3 Fig.1
Is Figure 1 newly created by the author? Is it a reprint of government materials? Write it so that the reader can understand it.
p.3 l.138 etc.
The position of some page breaks is not good.
p.7 Fig.2
Why explain only the waste lightings? Is it more important than other products? Please explain the transition of WEEE other than lighting equipment.
p.9 Table 8
What is Content Level? Explain it to the reader.
Author Response
Q1. This manuscript only introduces the measures of the Taiwan government, and does not suggest the academic significance of social science or the improvement of the system. It will be necessary to explain what is different from the text of the government policy introduction. Authors should also consider and suggest future improvements to WEEE.
Reply: As suggested by the reviewer, the description about the recycling efforts by the non-government groups in the Introduction has been added. Also, the author addressed some suggestions for the improvements of WEEE in the Conclusions.
“In addition, another resource recycling system in Taiwan for usable yet outdated products was conducted by the civil non-profit organizations that may be not registered as the recycling enterprises. Among them, the Tzu Chi, one of the Buddhist groups, may be the most famous non-profit recycling system.” (in the Introduction)
“To raise the WEEE recycling performance, the Taiwan EPA shall adopt further promotional measures, including the regulated WEEE items increased, the recycling fee rate of the green-mark ICT products enhanced, the planning of differential subsidy fee improved, and the expanded scale of recycling enterprises by merge.” (in the Conclusion)
Q2. p.3 l.120
What is the 4-in-1 Recycling program? What does 4 or 1 represent? For example, what part is better than overseas accuracy?
Reply: As suggested by the reviewer, the description about the 4-in-1 Recycling program in Taiwan has been enhanced to make it clear.
“In order to echo with the international trends regarding the sustainable waste management and the extended producer responsibility (EPR) [16, 33], the central competent authority (i.e., Environmental Protection Administration, abbreviated as EPA thereafter) in Taiwan promulgated the resource recycling system under the authorization of the Waste Management Act since 1998. More significantly, the EPA began promoting the 4-in-1 Program, which claimed the responsible enterprises to pay a recycling fee to the Recycling Management Fund. The Fund’s money will be mostly used as an incentive to integrate local governments, communities, and recycling enterprises for resource recycling promotion. Mainly due to its value for recycling and hazardous substances contained, the EPA continuously declared the regulated recyclable waste items. The status of WEEE recycling in Taiwan has been reviewed by the previous study [30] and other reports [34, 35]. These recycled WEEE articles were carried out by the registered recycling enterprises, which can be subsidized by the special recycling fund.”
Q3. Is Figure 1 newly created by the author? Is it a reprint of government materials? Write it so that the reader can understand it.
Reply: Based on the Re. [41], the flowchart in Fig.1 was prepared by the author.
Q4. p.3 l.138 etc.
The position of some page breaks is not good.
Reply: As suggested by the reviewer, the position of some page has been rearranged to make it sequential.
Q5. p.7 Fig.2
Why explain only the waste lightings? Is it more important than other products? Please explain the transition of WEEE other than lighting equipment.
Reply: Due to the official declaration based on the unit of set before 2016, it was not available for the statistics on the amounts of recycled home electrical & electronic appliances and IT products by weight except for waste lightings. (It has been explained in the first paragraph of the Sec. 3.2.)
Q6. p.9 Table 8
What is Content Level? Explain it to the reader.
Reply: As suggested by the reviewer, the term level in Table 8 has been explained to make it clear.
“The control level is officially announced by the central competent authority when handling the concentration of toxic chemical substance greater than the standard.”

Round 2
Reviewer 2 Report
In this manuscript, the author introduced the case study of the recycling of waste electrical and electronic equipment (WEEE) and its management of toxic substances in the Taiwan government.
I think the report has been revised a little to help readers.